# Emergence of OXA-48-like Carbapenemase-Producing *Escherichia coli* in Baranya County, Hungary

**DOI:** 10.3390/antibiotics15010044

**Published:** 2026-01-02

**Authors:** Fatma A. Mohamed, Mohamed Al-Bulushi, Szilvia Melegh, Bálint Timmer, Réka Meszéna, Csongor Freytag, Levente Laczkó, László Miló, Péter Urbán, Renáta Bőkényné-Tóth, Attila Gyenesei, Gábor Kardos, Adrienn Nyul, Edit Urbán, Tibor Pál, Ágnes Sonnevend

**Affiliations:** 1Department of Medical Microbiology, University of Pécs Medical School, 7624 Pécs, Hungary; fatmamohamed@pte.hu (F.A.M.); al.bulushi.mohamed@pte.hu (M.A.-B.); melegh.szilvia@pte.hu (S.M.); timmer.balint@pte.hu (B.T.); meszena.reka@pte.hu (R.M.); nyul.adrienn@pte.hu (A.N.); urban.edit@pte.hu (E.U.); pal.tibor2@pte.hu (T.P.); 2Department of Microbiology and Immunology, Faculty of Pharmacy, Zagazig University, Zagazig 44519, Egypt; 3Centre for Metagenomics, Multidisciplinary Health Industry Coordination Institute, University of Debrecen, 4032 Debrecen, Hungary; milo.laszlo@med.unideb.hu (L.M.); kg@med.unideb.hu (G.K.); 4Department of Bioinformatics, One Health Institute, Faculty of Health Sciences, University of Debrecen, 4032 Debrecen, Hungary; freytag.csongor@etk.unideb.hu (C.F.); laczko.levente@etk.unideb.hu (L.L.); 5HUN-REN-UD Conservation Biology Research Group, University of Debrecen, 4032 Debrecen, Hungary; 6Hungarian Centre of Genomics and Bioinformatics, Szentágothai Research Center, University of Pécs, 7624 Pécs, Hungary; urban.peter@pte.hu (P.U.); gyenesei.attila@pte.hu (A.G.); 7Department of Infection Control and Hospital Epidemiology, One Health Institute, Faculty of Health Sciences, University of Debrecen, 4032 Debrecen, Hungary; btoth.renata@etk.unideb.hu; 8Department of Planetary Health, One Health Institute, Faculty of Health Sciences, University of Debrecen, 4032 Debrecen, Hungary; 9Molecular Medicine Research Group, Szentágothai Research Center, University of Pécs, 7624 Pécs, Hungary

**Keywords:** OXA-48-like carbapenemases, *Escherichia coli*, high-risk clones, IncFIB-IncFIC plasmid, Hungary

## Abstract

**Background**: Carbapenem-resistant *Escherichia coli* (CREC) producing OXA-48-like carbapenemase was first detected in Hungary in 2022. The aim of the present study was to characterize such strains isolated in 2022–2025 in Baranya County, Hungary. **Methods**: Antibiotic susceptibility and the whole-genome sequence (WGS) of *E. coli* isolates, identified as OXA-48-like carbapenemase producers using the CARBA-5 NG test, were established. The transferability of *bla*_OXA-48-like_ plasmids was tested by conjugation. **Results**: Of the 6722 non-repeat *E. coli* isolates, 6 produced an OXA-48-like carbapenemase. They exhibited variable resistance to ertapenem and were susceptible to imipenem and meropenem. WGS revealed that all OXA-48-like producer *E. coli* belonged to high-risk clones: two clonally related OXA-181-producer *E. coli* ST405 were isolated in Hospital A, three OXA-244-producing *E. coli* ST38 (two identical via cgMLST from Hospital B), and an OXA-48-producing *E. coli* ST69. The *bla*_OXA-48_ and *bla*_OXA-244_ genes were chromosomally located, while *bla*_OXA-181_ was on a non-conjugative IncFIB-IncFIC plasmid. So far, the *bla*_OXA-181_-bearing plasmid of this incompatibility type has only been described in Ghana, but all *bla*_OXA-48-like_ gene-carrying transposons in this study have already been identified in Europe and other continents. The *E. coli* ST38 isolates, showing close association based on core genome SNP distances to European and Qatari strains, belonged to Cluster A and harbored *bla*_CTX-M-27._ All but the *E. coli* ST69 isolate had cephalosporinase gene(s). **Conclusions**: This study describes small-scale intra-hospital transfers of OXA-48-like carbapenemase-producer *E. coli*. Interestingly, *E. coli* ST405 of Hungary carried *bla*_OXA-181_ on an IncFIB-IncFIC plasmid, which has only been reported from Africa so far.

## 1. Introduction

The World Health Organization (WHO) and the European Center for Disease Prevention and Control (ECDC) both recognize carbapenem-resistant *Enterobacteriaceae* (CRE) as an urgent threat to public health [1]. The ECDC estimated that, in 2020, 30.9% of the total burden in disability-adjusted life years (DALYs) was from infections with carbapenem-resistant bacteria; among them, the highest number of deaths was attributable to carbapenem-resistant *Klebsiella pneumoniae* (4076 deaths) [1]. At the same time, the rate of carbapenem-resistant *Escherichia coli* was below 1% [1]. Carbapenem resistance is most commonly due to the production of carbapenemases belonging to classes A, B, and D of beta-lactamases. OXA-type enzymes are serine-type ß-lactamases belonging to class D of these enzymes [2]. They usually exhibit limited activity against third- and fourth-generation cephalosporins and monobactams, and their carbapenemase activity can also be weaker than that of other classes, particularly class B metallo-beta-lactamases [3]. The class D carbapenemases of *Enterobacteriaceae* are relatively closely related to OXA-48-like enzymes, representing three clusters (A: OXA-48, OXA-244, and OXA-162; Cluster B: OXA-181, OXA-232, and OXA-484; Cluster C: OXA-204) [4]. They often cause low-level resistance to carbapenems in vitro, thus posing a diagnostic challenge, for which their prevalence is likely underestimated [5]. Furthermore, OXA-48-like carbapenemase-producing *Enterobacterales* infections managed with carbapenems may fail treatment [6].

The *bla*_OXA-48-like_ genes are commonly carried on plasmids or integrated within the chromosome by international high-risk clones of *Klebsiella pneumoniae* (e.g., ST11, ST14, ST15, ST147, ST231, ST405) and *E. coli* (e.g., ST38, ST69, ST131 and ST410) and less frequently by other species of the family [4]. Such clones and, consequently, these enzymes are widespread globally and hence are often the second or third most common carbapenemases encountered. They are particularly common in the Middle East, North-Africa, Sub-Saharan Africa, and the Indian subcontinent, while they are relatively rare in North and South America [4,5,7]. Nevertheless, within a larger region, e.g., a continent, considerable variations may exist between countries in the prevalence and dominant subtypes of OXA-48-like enzymes. International travel, population movement, and spreading clones could alter existing prevalences [4,5,7]. In 2021, the ECDC issued a rapid risk assessment on the spread of OXA-244 carbapenemase-producing *E. coli* in the European Union and the United Kingdom and emphasized the need for molecular surveillance of the spread on the continent [8]. In Hungary, OXA-162-producing *K. pneumoniae* ST15 [9] and OXA-48-producing *K. pneumoniae* ST395 [10] were already described in 2012 and 2014, respectively. But, so far, the only reports on *bla*_OXA-48-like_ carrying *E. coli* was a poster on fifty carbapenem resistant isolates, of which three were ST38–*bla*_OXA-244_, one was ST131–*bla*_OXA-244_, one was ST448-*bla*_OXA-244_, one was ST38-*bla*_OXA-48_, one was ST410-*bla*_OXA-181_, and one was ST457-*bla*_OXA-232_ producers [8], and a Europe-wide multicenter study encountered a single OXA-244 producer *E. coli* ST131 in Hungary in 2023 [9]. However, neither study reported on the isolates’ respective resistome and the genetic support of the carbapenemase genes.

Our aim was to provide a detailed characterization of *E. coli* isolates identified as OXA-48-like producers by the CARBA-5 NG (NG Biotech SAS, Guipry, France) test in the Laboratory of Clinical Microbiology and Hospital Hygiene of the University of Pécs Clinical Center.

## 2. Results

### 2.1. Clinical Data and Antibiotic Susceptibility of the OXA-48-like Carbapenemase Producer Escherichia coli Isolates

Of the 6722 non-repeat *E. coli* isolated in the Laboratory of Clinical Microbiology and Hospital Hygiene of the University of Pécs Clinical Center between January 2022 and July 2025, only 6 were identified as OXA-48-like producers, i.e., the incidence of such isolates was less than 0.1% (Table 1).

All six of them were resistant to piperacillin-tazobactam, but they exhibited variable susceptibility to third-generation cephalosporins, aztreonam, ertapenem, aminoglycosides, and ciprofloxacin. All of them had an MIC below the clinical susceptibility breakpoint to imipenem and meropenem, and all were susceptible to ceftazidime-avibactam, cefiderocol, colistin, tigecycline, and fosfomycin (Table 2).

### 2.2. Whole-Genome Analysis

Based on whole-genome sequencing, the isolates belonged to sequence types ST38 (*n* = 3), ST405 (*n* = 2), and ST69 (*n* = 1), i.e., all were members of high-risk *E. coli* clones. The three *E. coli* ST38 isolates carried *bla*_OXA-244_, and the *E. coli* ST69 isolate carried the *bla*_OXA-48_ gene on its chromosome, whereas the two *E. coli* ST405 strains harbored *bla*_OXA-181_ on IncFIB-IncFIC plasmids (Table 3). The quality scores of the raw reads and assemblies, as well as the assembly statistics, are shown in Appendix A.

Beyond the *bla*_OXA-48-like_ genes, WGS did not identify other known carbapenemase genes. The variations in the susceptibility profiles of the strains (Table 2) reflected the varying presence of different beta-lactamase, aminoglycoside, fluoroquinolone, and sulfonamide resistance genes identified (Table 3).

Based on their cgMLST, the two *E. coli* ST405 (EC43905 and EC47538), both isolated from patients treated in 2022 in the Intensive Care Unit of Hospital A, and two *E. coli* ST38 (EC47258 and EC48956), isolated from patients treated in 2023 in the Surgical Ward of Hospital B, were clustered together (Figure 1).

Although the third *E. coli* ST38 (EC54214) from Hospital C was more distant from the ST38 cluster, all three OXA-244 producer *E. coli* ST38 from our region belonged to cluster A by the ECDC definition [8], i.e., carried the *bla*_OXA-244_ carbapenemase gene on the chromosome and possessed *bla*_CTX-M-27_ carried by an IncFIB-IncFII plasmid. In all strains, the genetic surroundings of chromosomally inserted *bla*_OXA-244_ on an IS1 bracketed composite transposon were identical to those previously described in the Netherlands (Figure 2B). Furthermore, comparison of whole genomes available from NCBI and Enterobase of OXA-244-producing *E. coli* ST38 Cluster A isolates from France, Germany, Norway, the Netherlands, Poland, and Qatar using Snippy revealed a maximum core genome SNP distance of only 65 SNPs. Although the *E. coli* ST38 from Hospital B and C were comparatively distant, both Hospital B isolates EC47258 and EC48956, as well as the Hospital C isolate EC54214, had the closest match to strains from Poland (Figure 2A).

A BLAST search of the NCBI database revealed that the *bla*_CTX-M-27_ genes carrying approximately 148 kb of IncFIB-IncFII plasmids in the Hungarian isolates were similar (>99.7% nucleotide identity) to the *bla*_CTX-M-27_ genes carrying plasmids pRIVM_C019217_2 (CP087378) from the Netherlands, pKresCPE0314 (OX359164.1) from Norway, and PP596845.1 plasmid from the Czech Republic, all of which were from of OXA-244 producer *E. coli* ST38 strains (Figure 3).

Isolates EC47258 and EC48956 from Hospital B also carried another extended spectrum beta-lactamase gene: *bla*_CTX-M-15_ on a 110 kb phage-like plasmid typed as IncFIB (H89-PhagePlasmid). These plasmids were hybrid elements containing both phage-related genes and an FIB replicon. A comparative analysis of these phage-plasmids against sequences in the NCBI database revealed that they are highly similar to plasmids harbored by a human *E. coli* isolate from the United Kingdom (MW590712.1) and those harbored by an environmental *E. coli* isolate from Switzerland (CP124485.1) and to pECOH89 (Accession number HG530657) from Germany, all of which carried an IS*Ecp1-bla*_CTX-M-15_ transposition unit (Figure 4).

The genetic surroundings of the chromosomally located *bla*_OXA-48_ gene in EC51489 *E. coli* ST69 were 99.9% identical to a 10,405 bp region of *Tn6237* (HG977710), comprising a ∆Tn1999.2 and plasmid-derived sequences (Figure 5).

The genetically related *E. coli* ST405 isolates EC43905 and EC47538 from Hospital A carried a different allele of OXA carbapenemase genes: *bla*_OXA-181._ This carbapenemase gene was located on an IncFIB-IncFIC plasmid, and its genetic surrounding was identical to those of *bla*_OXA-181_ carried on IncX3 plasmids found in the United Arab Emirates, Canada, and China and to an IncFIC plasmid present in a Ghanaian *E. coli* ST940 isolate (Figure 6).

### 2.3. In Vitro Transferability of OXA-48-like Carbapenemases

While attempts were made to conjugally transfer OXA-48-like carbapenemases, they were unsuccessful, even with those isolates in which the *bla*_OXA-48-like_ genes were plasmid-borne. Nevertheless, the *bla*_OXA-181_-carrying IncFIB-IncFIC plasmids were predicted to be conjugative by MOB-suite [11]; therefore, the failed conjugation could be the result of the low expression of conjugation genes or suboptimal in vitro experimental conditions.

## 3. Discussion

OXA-48-like carbapenemases are among the most common carbapenemases worldwide, with high prevalence in Europe [12]. In the 211 carbapenem-resistant *E. coli* isolates collected in Europe during the Carbapenem- and Colistin-Resistant Enterobacterales (CCRE) survey, *bla*_OXA-48-like_ was one of the second most frequently detected carbapenemase genes, accounting for 32% [12]. Although mostly present in *K. pneumoniae*, this type of carbapenemase has also been described in *E. coli* and other members of the *Enterobacterales* order, but in much lower frequency [5].

In Hungary, the first OXA-48-like carbapenemase-producing enteric bacterium, an OXA-162-producing *K. pneumoniae*, was reported in 2012 [9], followed by the emergence and interhospital spread of the OXA-48-producing *K. pneumoniae* ST395 clone in 2014 [10]. In both instances, cross-border transfer from Romania [9] or Ukraine [10] was assumed. So far, the only reports from Hungary on *bla*_OXA-48-like_ carrying *E. coli* were a poster on 50 carbapenem-resistant isolates [13] and a Europe-wide multicenter study encountering a single OXA-244 producer *E. coli* ST131 in Hungary in 2023 [14]. However, neither study provided a detailed characterization of the isolates with their respective resistome and the localization of the carbapenemase genes.

In our study, we demonstrated the transmission of OXA-244-producing *E. coli* ST38 and OXA-181-producing *E. coli* ST405 within respective hospital wards and the emergence of OXA-244-producing *E. coli* ST38 and OXA-48-producing ST69 in a third hospital. The occurrence of OXA-244-producing *E. coli* ST38 in Hospital C was supposedly an independent event, as this isolate did not cluster by cgMLST with those from Hospital B. No OXA-carbapenemase-producing *E. coli* was isolated after December 2023, while their rate in Hungary quadrupled (from 16.7% in 2023 to 66.7% in the first half of 2025—Dr. Ákos Tóth, Microbiological Reference Laboratory, National Center of Public Health and Pharmacy, Budapest—*personal communication*). It is tempting to speculate that the local disappearance of such strains could have been attributed to the increased cleaning and infection control measures introduced. However, while these could have been effective indeed in containing the spread and in the disappearance of the existing isolates, it was not accompanied by a general decrease in other CRE isolates locally. Comparing the periods of 2022–23 to 2024–July 2025, their rate actually increased from 1.1% to 2.2%. Nevertheless, during the study period, OXA-carbapenemase producer *Enterobacterales* remained rare in our setting, with only two OXA-48-like and NDM co-producing *K. pneumoniae* isolated in 2024 (unpublished observation).

The complete genomes of the isolates revealed that all three *E. coli* ST38 isolates carried *bla*_OXA-244_ on the chromosome and *bla*_CTX-M-27_ on an IncFIB-IncFIC plasmid. Based on this, our *E. coli* ST38 isolates belonged to Cluster A of this high-risk clone [8]. Comparison of the core genome SNP distances between our isolates and other European, as well as one Qatari, OXA-244-producing *E. coli* ST38 of Cluster A revealed high genetic similarity. Further to this, the *bla*_CTX-M-27_-carrying plasmids were also highly similar to plasmids carried by Norwegian, Czech, and Dutch *E. coli* ST38 Cluster A isolates [15,16]. All these suggest that a common source is still unknown. However, shedding light on this could be rather challenging, as it was shown in Qatar that none of the carriers of OXA-244 producer *E. coli* ST38 had any international travel history [17]. Interestingly, the OXA-244-producing *E. coli* ST38 of Hospital B also carried a second ESBL plasmid, a *bla*_CTX-M-15_-bearing IncFIB (H89-PhagePlasmid) phage-plasmid hybrid episome, which was highly similar to plasmids recovered in Germany from an *E. coli* clinical isolate [18], in Switzerland from a freshwater *E. coli* isolate, and from a traveler returning from South Asia to the United Kingdom [19], i.e., across a wide geographical area.

We also identified an OXA-48 producer *E. coli* ST69 in the present study. *E. coli* ST69 is a well-recognized high-risk clone, mostly producing ESBL enzymes [20]. However, isolates carrying class D carbepenemases, e.g., *bla*_OXA-244_, have also been reported from Denmark, Switzerland, France, and Poland [21,22]. Furthermore, an OXA-48 producer *E. coli* ST69 was described from Pakistan [23]. Of note is that the OXA-48 producer *E. coli* ST69 identified in our study did not possess any extended-spectrum beta-lactamase genes beyond the chromosomally located *bla*_OXA-48_, which remained susceptible to third-generation cephalosporins and carbapenems (Table 2) and was only recognized as a carbapenemase producer because its meropenem disc inhibition zone was below the screening cut-off value.

Further to the chromosomally located class D carbapenemases, a pair of OXA-181-producing *E. coli* ST405 was identified from Hospital A. Their cgMLST clustering suggested healthcare-associated transmission within the hospital. *E. coli* ST405 is typically associated with *bla*_NDM-5_ and has been identified in several European countries, including Finland, Denmark, France, the Netherlands, Ireland, and Sweden [24]. However, the ST405 strain in our study harbored *bla*_OXA-181_, a variant of OXA-48 endemic in India, where it was first discovered, and in Africa [25]. This may suggest that the strain was likely transferred through international travel; however, the index patient in Hospital A did not have any international travel history. Furthermore, the *bla*_OXA-181_ in these strains was located on an IS26 pseudo-composite transposon [26] also containing the *qnrS1* gene, which is identical to that found in IncX3 plasmids and carried by an IncFIB-FIC type plasmid. This plasmid has only been identified so far to carry *bla*_OXA-181_ in an enterotoxigenic *E. coli* ST940 from Ghana [27]. Although the *bla*_OXA-181_ was located on the same transposon, the plasmid, despite belonging to the same incompatibility type, did not show high similarity to p43905-1 and p47538-1, i.e., to the *bla*_OXA-181_-carrying plasmids of the current study.

Although four OXA-48-like carbapenemase genes identified were chromosomally coded and the two plasmid-borne ones could not be transferred by conjugation in vitro, all *bla*_OXA-48-like_ genes in this study were located on mobile genetic elements, which may facilitate their horizontal transfer. Furthermore, all OXA-48-like producer *E. coli* belonged to international high-risk clones, and despite the successful containment of nosocomial transmissions within two hospitals of two distinct clones (ST405 and ST38), these clones are recognized for their ability to successfully disseminate.

These observations support the view that clinical microbiology laboratories should carefully evaluate isolates showing carbapenem susceptibility below the screening cut-off to detect these difficult-to-recognize carbapenemase-producing isolates. Moreover, continuous whole-genome-based surveillance should be used to track and trace them in order to provide data to help break the transmission chain, at least within healthcare facilities.

## 4. Materials and Methods

**Bacterial Isolates**: Enterobacterial isolates exhibiting an inhibition zone of <28 mm with a 10 µg meropenem disc are routinely tested for their carbapenemase production using the mCIM test [28] in the Laboratory of Clinical Microbiology and Hospital Hygiene of the Department of Microbiology of the Clinical Center of the University of Pécs. If the mCIM test is positive, the type of carbapenemase produced is determined using NG-Test CARBA 5 (NG Biotech, Guipry, France). From 2022 until July 2025, six *Escherichia coli* isolates were identified by this algorithm as OXA-48-like carbapenemase producers from Hospitals A and B (secondary care hospitals with 500 and 300 beds, respectively) and Hospital C (tertiary care university hospital with 1400 beds) (listed in Table 1). The three hospitals are located in three different cities of Baranya County at least 40 kms apart.

The isolates’ species identification was performed using matrix-assisted laser desorption ionization–time-of-flight mass spectrometry (MALDI-TOF MS) (Bruker Daltonics, Bremen, Germany). Strains were stored at −80 °C in Tryptic Soy Broth (TSB, MAST, Merseyside, UK) containing 20% glycerol until further investigations.

**Antibiotic Susceptibility Testing**: The susceptibility to amikacin, amoxicillin/clavulanic acid, aztreonam, cefotaxime, ceftazidime/avibactam, ceftolozane/tazobactam, ciprofloxacin, colistin, ertapenem, gentamicin, imipenem, meropenem, piperacillin/tazobactam, tigecycline, tobramycin, and trimethoprim/sulfamethoxazole was established by broth microdilution using the SensiTitre DKMGN Plate (ThermoFisher Diagnostics, Landsmeer, The Netherlands), and the susceptibility to cefiderocol and fosfomycin (BioRad, Marnes-la-Coquette, France) was determined by disc diffusion, with *E. coli* ATCC25922 used as the control. All results were interpreted using EUCAST clinical breakpoints [29].

**Whole-Genome Sequencing and Analysis**: The complete genomes of strains identified as OXA-48-like producer *E. coli* were determined by 150 bp paired-end sequencing on the Illumina NovaSeq platform (San Diego, CA, USA) and long-read sequencing on Oxford Nanopore MinION (Oxford, UK) or on PacBio platforms (Menlo Park, CA, USA). Short- and long-read quality was assessed by FastQC and LongReadSum, respectively [30,31]. Hybrid assembly of short and long reads was performed using Unicycler v.0.5.0 [32]. The assembly quality was checked using QUAST [33], Samtools [34], Kraken2 and Braken [35], and CheckM [36].

MLST based on the Achtmann scheme [37] and core genome MLST of the six isolates were compared using the Ridom SeqSphere^®^ software v10.5.

Plasmid replicon types and resistance and virulence genes were identified with ABRicate (https://github.com/tseemann/abricate (accessed on 19 August 2025)) using PlasmidFinder [38], CARD, and VFDB databases [39,40], with the identity threshold set as a minimum of 95%. The transferability of plasmids was in silico assessed using MOB-suite [11]. The NCBI database was searched by nucleotide BLAST (https://blast.ncbi.nlm.nih.gov/Blast.cgi?PROGRAM=blastn&BLAST_PROGRAMS=megaBlast&PAGE_TYPE=BlastSearch (accessed on 15 September 2025)) to identify plasmids that are highly similar (>98% identity and 92% query coverage) to beta-lactamase-harboring plasmids of our isolates. Subsequently, the circular comparison of the plasmids was visualized by Proksee (https://proksee.ca/ (accessed on 14 October 2025)) [41]. Annotation was performed with Bakta v1.9.1. [42]. Linear comparison was performed with Clinker [43]. Prophage identification was detected by Phigaro, an integrated tool in Proksee. MobileOG-db [44] was used for mobile genetic element prediction as an integrated tool in Proksee.

Whole genomes of OXA-244-producing *E. coli* ST38 isolates available in the NCBI database and in Enterobase (*n* = 41) were downloaded from the respective databases (accessed on 13 August 2025), and SNP determination using the core genomes was analyzed using kSNP3 (v3.1). The single-nucleotide polymorphisms (SNPs) were calculated by reference free k-mer–based SNP difference matrix. The core genomes’ SNPs were used to calculate pairwise SNP distances between isolates to build a maximum likelihood phylogenetic tree. The tree was constructed by FastTree 2 and was visualized via Interactive Tree of Life, iTOL (https://itol.embl.de/ (accessed on 19 August 2025)).

**Conjugation**: *bla*_OXA-48-like_-carrying plasmids’ transferability was tested in mating-out assays using a sodium-azide-resistant derivative of rifampicin-resistant *E. coli* J53 (J53_RAZ_) as recipients. Conjugation was attempted at 30 °C and at 37 °C. Selection for transconjugants was carried out on a Tryptic Soy Agar containing 0.25 mg/L^−1^ of ertapenem and 150 mg/L^−1^ of sodium-azide [45].

## 5. Conclusions

The present study is the first to characterize OXA-48-like carbapenemase-producing *E. coli* isolated in Hungary by establishing their complete genomes, which allowed demonstration of clonal relationships and the similarity of class D carbapenemase-carrying transposons to isolates from different continents, even though the patients did not have any travel history. Furthermore, small-scale intra-hospital transfers and their successful containment were also revealed. As OXA-48-like carbapenemase-producing *Enterobacterales* are endemic in Germany, France, and Belgium and caused hospital outbreaks in Norway, Denmark, Czechia, and Poland, heightened awareness to identify such isolates, especially the OXA-244 producers with lower carbapenem MICs, is required in our region as well.

## Figures and Tables

**Figure 1 antibiotics-15-00044-f001:**
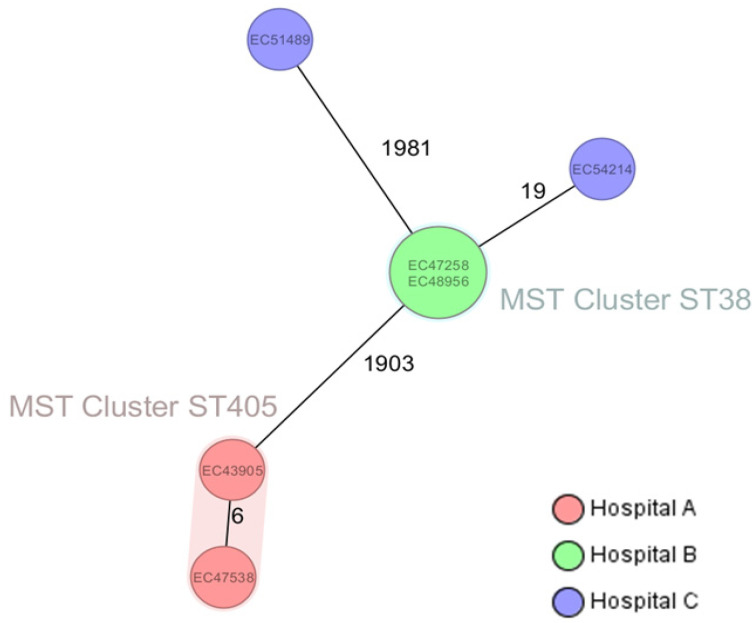
Ridom SeqSphere + MST of isolates of this study; logarithmic scale distance based on columns from *E. coli* cgMLST (2513); MST Cluster distance threshold: 10.

**Figure 2 antibiotics-15-00044-f002:**
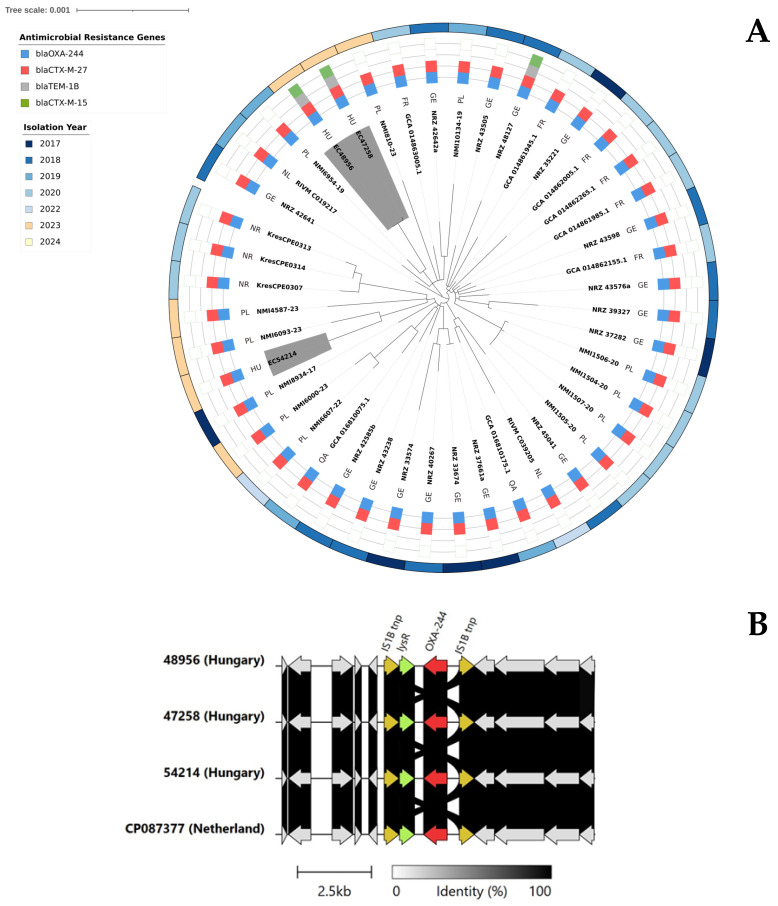
(**A**) Global phylogenomic analysis of 41 *E. coli* ST38 genomes from the EnteroBase and NCBI database. Isolates are colored by antimicrobial-resistant genes present and the year of isolation. The three Hungarian isolates identified in this study are highlighted in grey. FR, France; GE, Germany; HU, Hungary; NL, the Netherlands; NR, Norway; PL, Poland; QA, Qatar. (**B**) Comparison of the chromosomal region containing the *bla*_OXA-244_ gene in our Hungarian isolates with an identical selected one from the Netherlands, retrieved from NCBI databases. Color code: red, OXA-carbapenemase; orange, transposases; light green, *lysR*; grey, other genes.

**Figure 3 antibiotics-15-00044-f003:**
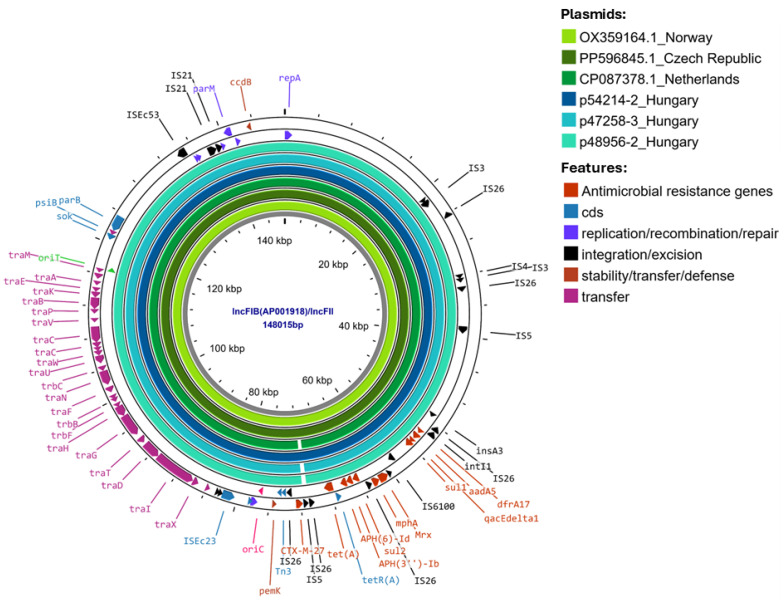
BLASTn comparison of IncFIB/IncFII plasmids p48956_2 (147,410 bp), p47258-4 (147,397 bp), and p54214_4 (150,133 bp) containing CTX-M-27 with plasmids retrieved from NCBI: pRIVM_C019217_2 (CP087378) from the Netherlands, PP5896845.1 from the Czech Republic, and pKresCPE0314 (OX359164.1) from Norway (used as a reference).

**Figure 4 antibiotics-15-00044-f004:**
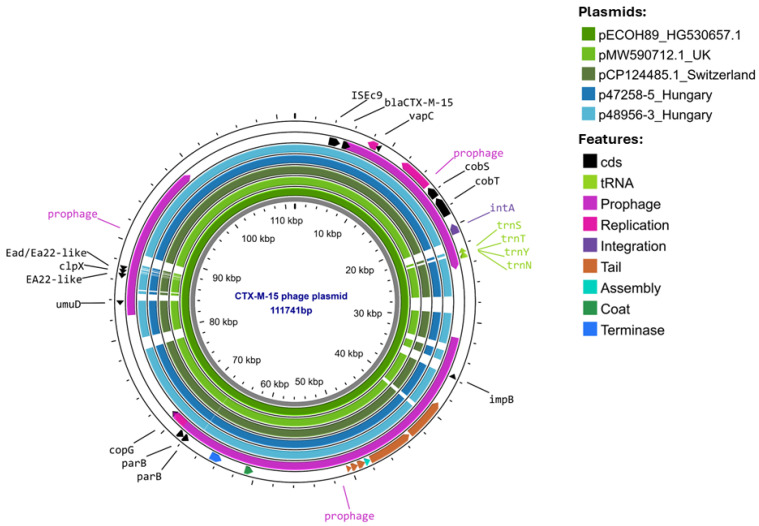
BLASTn comparison of phage plasmid harboring the *bla*_CTX-M-15_ gene to phage plasmids from Germany, the United Kingdom, and Switzerland; retrieved from NCBI.

**Figure 5 antibiotics-15-00044-f005:**
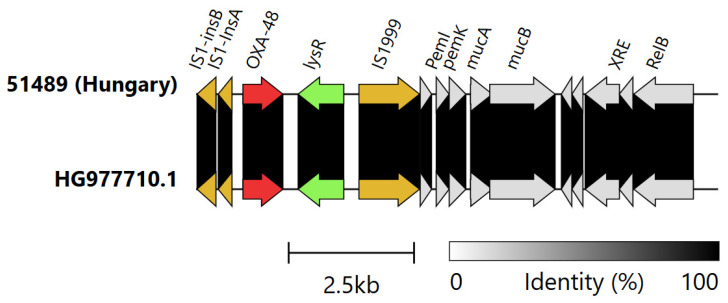
Genetic context of chromosomally located *bla*_OXA-48_ in the Hungarian *E. coli* ST69 compared to the Tn*6237* transposon. Color code: red, OXA-carbapenemase; orange, transposases; light green, *lysR*; grey, other genes.

**Figure 6 antibiotics-15-00044-f006:**
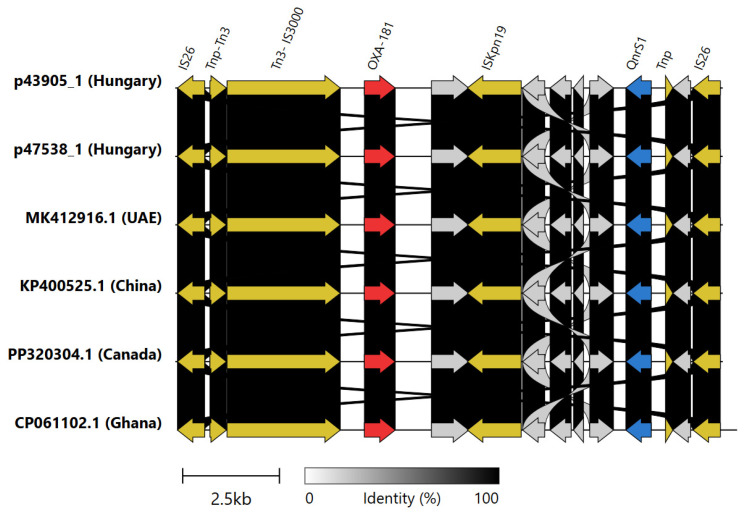
The genetic surroundings of the *bla*_OXA-181_ gene on IncFIB-IncFIC plasmids in the Hungarian *E. coli* ST405 strains compared to *bla*_OXA-181_ located on IncX3 plasmids MK412916, PP320304, KP400525, and IncFIC plasmid CP061102. Color code: red, OXA-carbapenemase; orange, transposases; blue, *qnrS1*; grey, other genes.

**Table 1 antibiotics-15-00044-t001:** Characteristics of OXA-48-like carbapenemase producer *Escherichia coli* isolates.

Isolate	Specimen	Date of Isolation	Ward/Hospital	Patient’s Age (Year)	Patient’s Gender	Colonization/ Infection
EC43905	Urine	24 October 2022	ICU-Hospital A	52	male	infection
EC47538	Urine	17 November 2022	ICU-Hospital A	75	female	infection
EC47258	Wound swab	26 October 2023	Surgery-Hospital B	79	male	infection
EC48956	Wound swab	7 November 2023	Surgery-Hospital B	50	male	colonization
EC54214	Urine	9 December 2023	Urology-Hospital C	53	male	infection
EC51489	Urine	23 November 2023	Transplant Surgery-Hospital C	32	female	infection

**Table 2 antibiotics-15-00044-t002:** Antibiotic susceptibility of the six OXA-48-like carbapenemase-producing *E. coli* identified in this study.

Lab Code	MIC (mg/L)																	Disc Diffusion Diameter (mm)	
	Amoxicillin/Clavulanic Acid	Piperacillin/Tazobactam	Cefotaxime	Ceftazidime	Ceftazidime/Avibactam	Ceftolozane/Tazobactam	Meropenem	Imipenem	Ertapenem	Aztreonam	Gentamicin	Amikacin	Tobramycin	Ciprofloxacin	Colistin	Tigecycline	Trimethoprim- Sulfamethoxazole	Fosfomycin 200 μg	Cefiderocol 30 μg
EC43905	>64	>32	>8	>16	**2**	>32	**1**	**1**	>2	>32	>8	**≤4**	4	>2	**1**	**≤0.25**	>8	**32**	**23**
EC47538	>64	>32	>8	>16	**2**	>16	**1**	**≤0.5**	>2	>32	>8	**≤4**	4	>2	**0.5**	**≤0.25**	>8	**35**	**23**
EC48956	>64	>32	>8	>16	**≤0.5**	8	**0.25**	**≤0.5**	1	>32	**≤0.5**	**≤4**	**≤1**	**≤0.06**	**0.5**	**≤0.25**	>8	**34**	**25**
EC47258	>64	>32	>8	16	**≤0.5**	8	**0.5**	**≤0.5**	2	32	**≤0.5**	**≤4**	**≤1**	**≤0.06**	**0.5**	**≤0.25**	>8	**33**	**25**
EC54214	>64	>32	>8	8	**≤0.5**	4	**0.5**	**≤0.5**	>2	16	**≤0.5**	**≤4**	**≤1**	**≤0.06**	**0.5**	**≤0.25**	>8	**34**	**25**
EC51489	>64	>32	**1** ^1^	**≤0.5**	**≤0.5**	**≤1**	**≤0.12**	**1**	**0.5**	**≤0.5**	**≤0.5**	**≤4**	**≤1**	**≤0.06**	**0.5**	**≤0.25**	**≤1**	**31**	**29**

^1^ Values written in bold indicate susceptibility.

**Table 3 antibiotics-15-00044-t003:** Characteristics of OXA-48-like carbapenemase-producing *E. coli* isolates based on their whole-genome sequences.

Lab Code	MLST	Plasmid/ Chromosome	Plasmid Incompatibility Type	Size (bp)	Antibiotic Resistance Genes
EC43905	ST405	Chromosome	NA	5,101,385	*sul2*, *aph(3″)-Ib*, *aph(6)-Id*, *tet(A)*, *floR*, *bla*_CTX-M-15_
		p43905-1	IncFIB-IncFIC	131,672	*bla*_OXA-181_, *qnrS1*, *erm(B)*, *mph(A)*, *bla_TEM-1B_*, *tet(B)*
		p43905-2	IncY	101,427	*mph(A)*, *aac(3)-IIa*, *dfrA17*, *aadA5*, *sul1*
		p43905-3	IncI(γ)	38,399	*bla* _CMY-42_
		p43905-4	ColpEC648	4187	None
EC47538	ST405	Chromosome	NA	5,101,050	*sul2*, *aph(3″)-Ib*, *aph(6)-Id*, *tet(A)*, *floR*, *bla*_CTX-M-15_
		p47538-1	IncFIB-IncFIC	131,642	*bla*_OXA-181_, *qnrS1*, *erm(B)*, *mph(A)*, *bla*_TEM-1B_, *tet(B)*
		p47538-2	IncY	101,312	*mph(A)*, *aac(3)-IIa*, *dfrA17*, *aadA5*, *sul1*
		p47538-3	IncI(γ)	38,399	*bla* _CMY-42_
		p47538-4	ColpEC648	4072	None
EC48956	ST38	Chromosome	NA	5,269,917	*bla* _OXA-244_
		p48956-1	IncFII	211,290	None
		p48956-2	IncFIB(AP001918)-IncFII	147,410	*bla*_CTX-M-27_, *tet(A)*, *aph(6)-Id*, *aadA5*, *aph(3″)-Ib*, *mph(A)*, *sul1*, *sul2*, *dfrA17*
		p48956-3	IncFIB(H89-PhagePlasmid)	110,850	*bla* _CTX-M-15_
		p48956-4	IncFII(pHN7A8)	64,449	*mph(A)*, *bla*_TEM-1B_
EC47258	ST38	Chromosome	NA	5,260,016	*bla* _OXA-244_
		p47258-1	IncFII	211,283	None
		p47258-2	IncFII(pHN7A8)	64,449	*mph(A)*, *bla*_TEM-1B_
		p47258-3	IncFIB(AP001918)-IncFII	147,397	*bla*_CTX-M-27_, *tet(A)*, *aph(6)-Id*, *aadA5*, *aph(3″)-Ib*, *mph(A)*, *sul1*, *sul2*, *dfrA17*
		p47258-4	IncFIB(H89-PhagePlasmid)	110,844	*bla* _CTX-M-15_
EC54214	ST38	Chromosome	NA	5,182,487	*bla* _OXA-244_
		p54214-1	IncFII	212,086	None
		p54214-2	IncFIB(AP001918)-IncFII	150,133	*bla*_CTX-M-27_, *tet(A)*, *aph(6)-Id*, *aadA5*, *aph(3″)-Ib*, *mph(A)*, *sul1*, *sul2*, *dfrA17*
		p54214-3	IncFIB(pLF82-PhagePlasmid)	111,076	None
EC51489	ST69	Chromosome	NA	5,194,121	*bla* _OXA-48_
		p51489-1	IncFIC(FII)-IncFIB(AP001918)-IncFIA	139,720	*bla*_TEM-1B_, *tet(A)*, *aph(6)-Id*, *aph(3″)-Ib*, *sul2*
		p51489-2	NT	3779	None

NA: Not applicable; NT: typable.

## Data Availability

The original data presented in the study are openly available in NCBI under the following: Bioproject: PRJNA1302334; Biosamples: SAMN50488977-SAMN50488980, SAMN50468512, and SAMN50468485; GenBank accession numbers: JBTANG000000000, JBQIAW000000000, JBRYHV000000000, JBQIAX000000000, JBQIAY000000000, and JBQIAZ000000000.

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
