# Peer review of "Emergence of OXA-48-like Carbapenemase-Producing Escherichia coli in Baranya County, Hungary"

_antibiotics, 2026, doi:10.3390/antibiotics15010044_

Round 1
Reviewer 1 Report
Comments and Suggestions for Authors
Interesting paper with detailed genomic insight into the OXA-48 producing E. coli in the Baranya County, Hungary.
I think the abstract can be shortened (the results are too detailed for the abstract). The discussion can also be improved based on the results. For example, there is no mention of the high risk clones in the results, yet the conclusion mentions HR clones.
Figure 2B is too small, please enlarge to make it clearer for the readers to see.
Can you reflect more on the failed conjugation, and possible reasons for that? How does the genomic data relate to that failure (if any markers for transferability).
Given the low prevalence of OXA-48 in E. coli, can you comment on the overall prevalence in your setting in other organisms?
L255: "six" is a typo.
Overall it's an interesting study showing high similarity to other parts of the world.
Reviewer 2 Report
Comments and Suggestions for Authors
The paper “Emergence of OXA-48-like carbapenemase-producing Escherichia coli in Baranya County, Hungary ” describes the characteristics of strains isolated in 2022-2025 in Baranya County, Hungary. Also tell the antibiotic susceptibility, and whole genome sequence (WGS) of E. coli isolates, identified as OXA-48-like carbapenemase producers
The authors isolated 6722 non-repeat E. coli isolates; six produced an OXA-48-like car-30 bapenemase. They found small-scale intra-hospital transfers of OXA-48-like car-42 bapenemase producer E. coli clones.
The manuscript contains significant information and has several areas for improvement, as described below.
All results should be better and more discussed
Methods should be better described and in detail, describing the controls
The authors used strains from patients. Informed consent forms must be shown and described.
The project's registration with a bioethics committee must be described.
Phylogenetic analyses are suggested using a model that fits the tree for a better description, analysis, and discussion of the strain sequences and epidemiology. It is recommended, if possible, to include strains from near the region or some of medical interest to determine and discuss their origin.
In the results section, it is important to describe the quality and integrity of the DNA samples, the quality parameters of the assemblies, and the sequence annotation.
Can the authors provide accession numbers to any databases containing the sequence data?
The results section must describe how the strains were obtained experimentally.
It is suggested to describe the general characteristics of the strain genomes, including size, GC content, and other characteristics, in addition to those described in the table.
Is there a correlation between the susceptibility tests and the results obtained from the sequences?
Apparently, six strains are reported. It is suggested that data for all strains be included in all analyses. In some cases, only some of them are present.
It is suggested that the quality of all figures be improved. In some, the text and some data are not visible.
Reviewer 3 Report
Comments and Suggestions for Authors
Fatma et al. described the genomic characteristics of six OXA-48-like carbapenemase-producing E. coli isolates from Hungary, and the study highlighted the importance of continuous, whole-genome-based surveillance to track and trace transmission chains within and among healthcare facilities. The manuscript is of high quality and well written. I only have a few minor comments and suggestions:
- Regarding the MLST typing, I am familiar with the Achtman and Pasteur schemes. What is the “Warwick profiles” mentioned in line 94? Is it the same as these two, or does it represent an independent scheme? It would be helpful for readers if the authors could clarify this in the manuscript.
- For Figure 2A, please provide the full names of the country abbreviations in the caption.
- For Figures 3 and 4, please group the legends.
- For the linear alignments shown in Figures 2, 5, and 6, is there any specific meaning associated with the colors of the genes? If so, please include this information in the caption.
Reviewer 4 Report
Comments and Suggestions for Authors
This article summarizes isolation of OXA-48-like carbapenemase producing carbapenem-resistant Escherichia coli between the years 2022-2025 in Baranya County, Hungary. On the six isolates available, the authors identified resistance to some of the third generation cephalosporins, and their WGS analyses revealed 2 of OXA-181-producing, 3 of OXA-244-producing, and 1 OXA-48-producing E. coli strains and identified their association to European and worldwide strains. The topic is very important and timely and will therefore attract attention of readers from various fields. Thus, the publication of this very important article is highly recommended. However, there are several minor issues that need to be addressed before the manuscript can be accepted for publication.
- Introduction is very concise and does not summarize the literature in depth enough to give the reader basic information on the carbapenem resistant strains. Especially, work on OXA-48-like carbapenemase producing carbapenem-resistant bacteria strains isolated in Europe and outside of Europe must be summarized in more depth.
- It is interesting that the last OXA-carbapenemase producer Enterobacterales isolated in November 2023, no OXA-carbapenemase producer was isolated in the past two years. Authors should comment on this finding and compare with worldwide incidences of OXA-carbapenemase producer within these two years.
- Conclusions must be briefly extended to comparisons with key findings of other European studies.
Round 2
Reviewer 2 Report
Comments and Suggestions for Authors
I reviewed the paper “Emergence of OXA-48-like carbapenemase-producing Escherichia coli in Baranya County, Hungary ”. The manuscript has improved significantly. Questions and comments were addressed. It is recommended for publication.